# Parallel Submodular Function Minimization

**Deeparnab Chakrabarty** [*]
Dartmouth College
Hanover, USA
deeparnab@dartmouth.edu

**Andrei Graur** [†]
Stanford University
Stanford, USA
agraur@stanford.edu

**Haotian Jiang**
Microsoft Research
Redmond, USA
jhtdavid96@gmail.com

**Aaron Sidford** [‡]
Stanford University
Stanford, USA
sidford@stanford.edu

## Abstract

We consider the parallel complexity of submodular function minimization (SFM). We provide a pair of methods which obtain two new query versus depth trade-offs a submodular function defined on subsets of $n$ elements that has integer values between $-M$ and $M$. The first method has depth 2 and query complexity $n^{O(M)}$ and the second method has depth $\widetilde{O}(n^{1/3}M^{2/3})$ and query complexity $O(\text{poly}(n, M))$. Despite a line of work on improved parallel lower bounds for SFM, prior to our work the only known algorithms for parallel SFM either followed from more general methods for sequential SFM or highly-parallel minimization of convex $\ell_2$-Lipschitz functions. Interestingly, to obtain our second result we provide the first highly-parallel algorithm for minimizing $\ell_\infty$-Lipschitz function over the hypercube which obtains near-optimal depth for obtaining constant accuracy.

## 1   Introduction

A function $f : 2^{[n]} \to \mathbb{Z}$ is *submodular* if it has the diminishing marginal return property that $f(S \cup \{i\}) - f(S) \geq f(T \cup \{i\}) - f(T)$ for all elements $i \in [n]$ and subsets $S \subseteq T \subseteq [n] \setminus \{i\}$, i.e., the function value increase of adding $i$ to a set $S$ is at least that of adding $i$ to a superset $T$ that does not contain $i$. Submodular functions are used to model a range of problems arising machine learning [KG05, KNTB09, KC10], operations research [FG88, QS95], and economics [Top98]. The *submodular function minimization (SFM) problem* consists of finding the subset $S$ with the smallest $f(S)$, given evaluation oracle access to the submodular function $f$, using as few queries as possible. SFM has numerous applications. For example, in natural language processing SFM has played a key role in speech analysis where [LB11, JB11] modeled the task of optimally selecting terse, yet diverse, training data out of a large speech dataset as a SFM problem; in computer vision, the task of energy minimization was reduced to SFM [BVZ01, KKT08, KT10, JBS13, KBJ+15].

Seminal work of [GLS81] established that SFM can be solved with a polynomial number of queries and in polynomial time. In the decades since, there has been extensive research characterizing the query and computational complexity of SFM. When $f$ is integer valued with values between

---

[*]Supported in part by NSF grant #2041920.

[†]Supported in party by NSF CAREER Award CCF-1844855, NSF Grant CCF-1955039, and Stanford Management Science & Engineering Department Nakagawa Fellowship

[‡]Supported in part by a Microsoft Research Faculty Fellowship, NSF CAREER Award CCF-1844855, NSF Grant CCF-1955039, a PayPal research award, and a Sloan Research Fellowship.

37th Conference on Neural Information Processing Systems (NeurIPS 2023).

$-M$ and $M$, the state-of-the-art query complexities are $\widetilde{O}(nM^2)$[4] due to [CLSW17] and [ALS20], $\widetilde{O}(n^2 \log M)$ and $\widetilde{O}(n^3)$, both due to [LSW15] (the latter query complexity being improved by a factor of $\log^2 n$ in [Jia21]), and $O(n^2 \log n)$ (with an exponential runtime) due to [Jia22]. The largest query complexity lower bound is $\Omega(n \log n)$ for deterministic algorithms due to [CGJS22].

The algorithms underlying the above results are highly *sequential*. State-of-the-art SFM algorithms use at least a linear number of rounds of queries to the evaluation oracle ($\widetilde{O}(nM^2)$ rounds in [ALS20] and $O(n \log n)$ rounds in [Jia22]). Although there is a trivial 1-round algorithm that queries all $2^n$ input of $f$ in parallel, all known polynomial time SFM algorithms use $\Omega(n)$ rounds.

With the prevalence of parallel computation in practice and applications of SFM to problems involving massive data sets, recently there has been an increasing number of works which study the *parallel* complexity of SFM, i.e., the smallest number of rounds a SFM algorithm must take. Starting with [BS20], significant progress has been made in proving *lower bounds* on the parallel complexity of SFM. It is now known that any SFM algorithm making polynomially many queries needs $\widetilde{\Omega}(n^{1/3})$ rounds when $M = \Theta(n)$ [CCK21] and $\widetilde{\Omega}(n)$ rounds when $M = \Theta(n^n)$ [CGJS22].

In contrast to the significant progress on lower bounds, less progress has been made on designing SFM algorithms with small parallel complexity. This is due in part to the established lower bounds; as noted above, if we desire *range-independent* parallel complexity bounds, i.e., independent of the range $M$, then it is impossible to obtain an $O(n^{1-\varepsilon})$-round SFM algorithm, for any constant $\varepsilon > 0$. The central question motivating this paper is what non-trivial parallel speedups for SFM are possible if we allow methods with a *range-dependent* parallel complexity.

> *Is it possible to obtain query-efficient $o(n)$-round $M$-dependent algorithms for SFM?*

In this paper we provide positive answers to the above question. Our first result is an algorithm that runs in $\widetilde{O}(n^{1/3}M^{2/3})$ rounds with query complexity $\widetilde{O}(n^2 M^2)$. To achieve this result, we first provide a generic reduction from parallel SFM to parallel convex optimization (a well studied problem discussed below). While naively this approach yields a sublinear bound of $\widetilde{O}(n^{2/3}M^{2/3})$ rounds, we show how to further improve these convex optimization methods in our setting.

Our second result is a simple 2-round SFM algorithm (Algorithm 1) with query complexity $n^{O(M)}$. For constant $M$, the parallel complexity of 2 is optimal among the class of query-efficient algorithms that query the minimizer [CLSW17]. It is instructive to contrast our second result to the lower bound in [CCK21], where it is proved that when $M = n$, any algorithm with query complexity $n^{M^{1-\delta}}$ for any constant $\delta > 0$ must proceed in $n^{\Omega(\delta)}$ rounds.

**Highly Parallel Convex Optimization.** Motivated by applications to distributed and large scale optimization [BPC+11, GR18], the question of highly parallel convex optimization has received significant attention in the last decade. Formally, the task is to find an (approximate) minimizer of a convex function this is Lipschitz in some norm, using as few rounds of $O(\mathsf{poly}(n))$ parallel queries to a subgradient oracle as possible. Over the past few years, there has been progress on both upper [DBW12, SBB+18, BJL+19] and lower bounds [Nem94, BS18, DG19] for this problem.

This line of work is particularly relevant as SFM reduces to minimizing the *Lovász extension* (see Fact 2.4) [Lov83] over $[0, 1]^n$, which is convex and $O(M)$-Lipschitz in $\ell_\infty$ (see Definition 2.1). In Section 2.1 we provide a straightforward reduction from this problem to unconstrained minimization of a $O(L)$-Lipschitz function in $\ell_\infty$ where the minimizer has $\ell_\infty$ norm $O(1)$. Consequently, improved parallel $\ell_\infty$-Lipschitz convex optimization algorithms can imply improved parallel SFM algorithms.

Many prior algorithms for highly parallel convex optimization focused on convex functions that are $\ell_2$-Lipschitz and some were written only for unconstrained optimization problems. Naively using these algorithms for $\ell_\infty$ yields parallel complexities of $\widetilde{O}(n^{3/4}/\varepsilon)$ [DBW12] and $\widetilde{O}(n^{2/3}/\varepsilon^{2/3})$ [BJL+19, CJJ+23][5] for convex functions that are $\ell_\infty$-Lipschitz.

Our first SFM result follows from an improved algorithm that we develop that finds $\varepsilon$-approximate minimizers for convex $\ell_\infty$-Lipschitz functions in $\widetilde{O}(n^{1/3}/\varepsilon^{2/3})$ rounds. Interestingly, for constant $\varepsilon > 0$, the dependence on $n$ in this improved parallel complexity is optimal up to logarithmic

---

[4]Throughout, we use $\widetilde{O}(f(n))$ to hide $\mathsf{poly}(\log f(n), \log n)$ factors.

[5][BJL+19, CJJ+23] study the $\ell_2$ setting. To obtain the corresponding $\widetilde{O}(n^{2/3}/\varepsilon^{2/3})$ result for $\ell_\infty$, we combine their result with Lemma 2.3 given in this paper.

factors; [Nem94] proved a lower bound of $\widetilde{\Omega}(n^{1/3} \ln(1/\varepsilon))$ on the round complexity of any algorithm obtaining an $\varepsilon$-approximate minimizer over $[0,1]^n$ for $\ell_\infty$-Lipschitz convex functions. For constant $\varepsilon > 0$, this lower bound also applies to unconstrained $\ell_\infty$-optimization [DG19].

| Paper | Year | Parallel Rounds | | Paper | Year | Parallel Rounds |
|---|---|---|---|---|---|---|
| [ALS20] | 2020 | $\widetilde{O}(nM^2)$ | | Subgradient Descent | 1960s | $O(n/\varepsilon^2)$ |
| [JLSW20] | 2020 | $\widetilde{O}(n \log M)$ | | Cutting Plane | 1965 | $\widetilde{O}(n \log(1/\varepsilon))$ |
| [Jia22] | 2021 | $\widetilde{O}(n)$ | | [BJL$^+$19, CJJ$^+$23]* | 2019 | $\widetilde{O}(n^{2/3}/\varepsilon^{2/3})$ |
| **This paper** | 2023 | $\widetilde{O}(n^{1/3}M^{2/3})$ | | **This paper** | 2023 | $\widetilde{O}(n^{1/3}/\varepsilon^{2/3})$ |

Table 1: State-of-the-art parallel complexity for SFM and $\ell_\infty$-optimization. See Section 1.2 for references on cutting plane methods. *The $\widetilde{O}(n^{2/3}/\varepsilon^{2/3})$ result uses Lemma 2.3 from this paper.

**Notation.** We let $[n] := \{1, \ldots, n\}$ for any $n \in \mathbb{Z}_{>0}$. We let $\mathbf{I}_n$ denote the identity matrix in $\mathbb{R}^{n \times n}$.

## 1.1 Problems, Results, and Approach

Here we formally define the problems we consider, present our results, and discuss some of the key insights of our approach. This section is organized as follows. We begin by presenting the parallel computation model. Then, we introduce the parallel $\ell_p$-Lipschitz convex optimization problem, which is closely tied to parallel SFM. We present our improvement for the $\ell_\infty$-Lipschitz setting and offer a brief overview of our techniques in obtaining this result. Finally, we formally introduce the SFM setup, along with the new results we obtain (Theorem 1.3 and Theorem 1.4).

**Parallel Complexity Model.** We consider the standard black-box query model for optimization, where we are given a convex function $f : \mathcal{D} \to \mathbb{R}$, with domain $\mathcal{D} \subset \mathbb{R}^n$, accessed through an oracle. Parallel algorithms for minimizing $f$ proceed in rounds, where in each round the algorithm can submit a set of queries to the oracle in parallel. The *parallel complexity* of an algorithm is the total number of rounds it uses and it captures how "sequential" the algorithm is. Additionally, we consider the *query complexity* of an algorithm which is the total number of queries it makes.

**Parallel $\ell_p$-Lipschitz Convex Optimization.** In Section 2 we consider the problem of minimizing a convex function $f : \mathbb{R}^n \to \mathbb{R}$ given access to a *subgradient oracle* $g : \mathbb{R}^n \to \mathbb{R}^n$, which when queried with point $x \in \mathbb{R}^n$ returns a vector $g(x)$ that is a subgradient, denote $g(x) \in \partial(f(x))$ where $\partial(f(x))$ is the set of all subgradients of $f$ at $x$, i.e., $v \in \partial f(x)$ if and only if $f(y) \geq f(x) + v^\top(y - x)$ for all $y \in \mathbb{R}^n$. Furthermore, we assume that $f$ is $L$-Lipschitz with respect to a norm $\|\cdot\|$.

**Definition 1.1** ($L$-Lipschitzness with respect to a given norm)**.** *We say that $f : \mathbb{R}^n \to \mathbb{R}$ is $L$-Lipschitz with respect to $\|\cdot\|$ for norm $\|\cdot\|$ if $|f(x) - f(y)| \leq L \|x - y\|$ for all $x, y \in \mathbb{R}^n$. If $f$ is $O(1)$-Lipschitz with respect to a norm $\|\cdot\|$, we say that $f$ is $\|\cdot\|$-Lipschitz. When the norm is $\|\cdot\|_p$ we alternatively say that $f$ is $L$-Lipschitz in $\ell_p$ and that $f$ is $\ell_p$-Lipschitz respectively.*

There is a broad line of work on studying the parallel complexity of $\ell_p$-Lipschitz convex optimization in which the goal is to efficiently compute an $\varepsilon$-approximate minimizer (i.e., a point $y$ with $f(y) \leq \inf_x f(x)$) of a $\ell_p$-Lipschitz convex function where the minimizer either has $\ell_p$-norm $O(1)$ or the problem is constrained to the $\ell_p$-norm ball of of radius $O(1)$. The case when $p = 2$ is perhaps the most well studied and our new result regarding $\ell_\infty$-Lipschitz convex optimization builds upon a result in [CJJ$^+$23], Theorem 1, which considers this setting. The statement of our result is below:

**Theorem 1.2** (Parallel Convex Optimization in $\ell_\infty$)**.** *There is an algorithm that when given a subgradient oracle for convex $f : \mathbb{R}^n \to \mathbb{R}$ that is 1-Lipschitz in $\ell_\infty$ and has a minimizer $x_*$ with $\|x_*\|_\infty \leq 1$ computes an $\varepsilon$-approximate minimizer of $f$ in $\widetilde{O}(n^{1/3}\varepsilon^{-2/3})$ rounds and $\widetilde{O}(n\varepsilon^{-2})$ queries.*

In fact, we obtain a more general result that solves stochastic variants of parallel $\ell_\infty$-optimization (see Theorem 2.2) and Theorem 1.2 is an important corollary of this more general result.

Our parallel convex optimization algorithms build on machinery developed for highly-parallel algorithms for minimizing convex functions that are $\ell_2$-Lipschitz. These methods consider a convolution of $f$ with a centered Gaussian with covariance $\rho^2 \mathbf{I}_n$ (also referred to as *Gaussian smoothing*), and then apply optimization methods [CJJ$^+$23, ACJ$^+$21, BJL$^+$19] to this smooth function. By leveraging

the properties of this smoothing and the convergence rate of the associated optimization methods, they obtain their parallel complexities. Since functions which are $\ell_\infty$-Lipschitz are also $\ell_2$-Lipschitz, the above algorithms also apply to $\ell_\infty$-Lipschitz functions but they give us a suboptimal dependence of $n^{2/3}$ on the dimension. We improve the dependence on dimension by utilizing the $\ell_\infty$-Lipschitzness of our function to add more Gaussian smoothing. This allows us to obtain a $n^{1/3}$ dependence on the dimension, which is optimal up to polylogarithmic factors for constant $\varepsilon$ [Nem94, DG19]. The key observation is that convolving an $\ell_\infty$-Lipschitz function with a Gaussian of covariance $\rho^2 \mathbf{I}_n$ changes the function value by no more than $O(\rho \sqrt{\log n})$ (see Lemma 2.6), whereas for $\ell_2$-Lipschitz functions it could change the function value by $O(\rho \sqrt{n})$.

**Submodular Function Minimization.** In SFM, we assume the submodular function $f : 2^{[n]} \to \mathbb{Z}$ is given by an *evaluation oracle*, which when queried with $S \subseteq [n]$, returns the value of $f(S)$. Throughout the paper, we assume that $f(S) \in [-M, M]$ for some $M \in \mathbb{Z}_{>0}$, and $f(\emptyset) = 0$. The assumption that $f(\emptyset) = 0$ can be made without loss of generality by instead minimizing $\tilde{f}(S) := f(S) - f(\emptyset)$; this transformation can be implemented with one query and moves the range of $f$ by at most $\pm M$, turning any dependence on $M$ in an algorithms' complexity to $2M$.

Note that this submodular function minimization setup is different from the setup of parallel convex optimization, as $f$ only defined on the vertices of the unit hypercube. Nonetheless, it is known that there is a convex function $f_{\mathsf{Lov}}$ defined on $[0,1]^n$, known as the Lovasz Extension, such that optimizing $f_{\mathsf{Lov}}$ suffices for optimizing $f$. Additionally, it is known how to compute a subgradient of $f_{\mathsf{Lov}}$ at any point $x$ in 1 round using at most $n$ evaluation queries to $f$ (as highlighted in Fact 2.4).

Now, we are ready to present our first result on parallel SFM. Later, in Section 2.1, we provide a more general version of this theorem, Theorem 2.5, which gives improved parallel complexities for approximately minimizing bounded, real-valued submodular functions.

**Theorem 1.3** (Sublinear Parallel SFM). *There is an algorithm that, when given an evaluation oracle for submodular $f : 2^{[n]} \to \mathbb{Z}$ with $f(\emptyset) = 0$ and $|f(S)| \leq M$ for all $S \subseteq [n]$, finds a minimizer of $f$ in $\widetilde{O}(n^{1/3}M^{2/3})$ rounds and $\widetilde{O}(n^2 M^2)$ queries.*

As discussed, Theorem 1.3, is obtained by using and enhancing tools for optimizing Lipschitz convex functions with a subgradient oracle. We in fact prove a more general result, namely that $\widetilde{O}(n^{1/3}/\varepsilon^{2/3})$ rounds and $\widetilde{O}(n^2/\varepsilon^2)$ queries are sufficient to find an $\varepsilon M$-approximate minimizer (Theorem 2.5). Since the function is integer valued, approximating the scaled Lovász extension to $\varepsilon \approx \Theta(1/M)$ gives the exact minimizer to the submodular function. Our proof of Theorem 2.5 follows from our new result on $\ell_\infty$-convex optimization (Theorem 1.2). By applying it to a scaled version of the Lovász extension; it is known that if $f$ is a submodular function with range $\mathbb{Z} \cap [-M, +M]$, then the Lovász extension scaled by $O(1/M)$ is a convex function which is $\ell_\infty$-Lipschitz. However, it is important to note that SFM is only equivalent to *constrained* minimization of the Lovász extension in $[0,1]^n$, while Theorem 1.2 below is *unconstrained* (e.g. applies for minimizing over $\mathbb{R}^n$). To apply Theorem 1.2 in the context of SFM, we give a general reduction from constrained to unconstrained optimization by adding a regularizer that restricts the minimizer of the regularized function to the constrainted set (see Lemma 2.3). This reduction is a generic technique and might be of independent utility.

**Parallel SFM in Two Rounds.** Our second SFM result is a simple combinatorial 2-round algorithm which is efficient for functions of constant range.

**Theorem 1.4** (Two-round Parallel SFM). *There is an algorithm (Algorithm 1) that when given an evaluation oracle for submodular $f : 2^{[n]} \to \mathbb{Z}$ with $f(\emptyset) = 0$ and $|f(S)| \leq M$ for all $S \subseteq [n]$ finds a minimizer of $f$ in 2 rounds and $O(n^{M+1})$ queries.*

The algorithm proving Theorem 1.4 relies on two key observations. First, if $S_*$ is the minimizer of maximum size, for every subset $T \subseteq S_*$ and $i \in [n]$ with $f(T \cup \{i\}) \leq f(T)$, we have $i \in S_*$. In other words, every element with a non-positive marginal at a subset $T \subseteq S^*$ is also contained in $S_*$. This leads to the idea of *augmenting* a set $T$ by the set $T' = T \cup \{i : f(T \cup \{i\}) \leq f(T)\}$. Secondly, every subgradient $g \in \partial f(x)$ has at most $M$ entries that are strictly positive. This ensures that there exists an $M$-sparse subset $T \subseteq S_*$ with the property that $f(T \cup \{i\}) \leq f(T)$ for all $i \in S_* \setminus T$. Consequently, our algorithm proceeds by augmenting all $M$-sparse sets, as it is guaranteed that one of these augmented sets is the maximal minimizer (see Section 3 for more details).

## 1.2 Related Work

**Parallel Convex Optimization.** As mentioned in Section 1.1, there are a number of works studying the parallel complexity of $\ell_p$-*Lipschitz convex optimization*. Perhaps, the most well-studied case is that of $p = 2$. In this case, the classic subgradient descent algorithm achieves a parallel complexity of $O(\varepsilon^{-2})$-rounds and the standard cutting-plane methods achieve a parallel complexity of $O(n \log(1/\varepsilon))$-rounds. [DBW12, BJL$^+$19, CJJ$^+$23] improved upon this rate, achieving parallel complexities of $\widetilde{O}(n^{1/4}\varepsilon^{-1})$ [DBW12] and $\widetilde{O}(n^{1/3}\epsilon^{-2/3})$ [BJL$^+$19, CJJ$^+$23] respectively. The implications of these results for the $p = \infty$ case, which is the object of study in our paper, were discussed earlier and we are unaware of works on alternative upper bounds for $p = \infty$.

In terms of lower bounds, the $p = \infty$ case was studied in the prescient paper of [Nem94] which obtains a $\widetilde{\Omega}(n^{1/3} \ln(1/\varepsilon))$ lower bound for minimizing $\ell_\infty$-Lipschitz functions over the $\ell_\infty$-ball (see also [DG19]). When $\varepsilon$ is a constant, our upper bound matches this lower bound, though our dependence on $\varepsilon$ is polynomial instead of logarithmic. For the $p = 2$ case, [Nem94, BS18] proved a tight lower bound of $\Omega(1/\varepsilon^2)$ on the parallel complexity when $1/\varepsilon^2 \leq \widetilde{O}(n^{1/3})$, i.e., the parallel complexity of subgradient descent is optimal up to $\widetilde{O}(n^{1/3})$ rounds of queries. This was later improved by [BJL$^+$19], which showed that subgradient descent is optimal up to $\widetilde{O}(n^{1/2})$ rounds. [DG19] considered the general $p$ case (and other non-Euclidean settings) and proved a lower bound of $\Omega(\varepsilon^{-p})$ lower bound on the parallel complexity for $2 \leq p < \infty$, $\Omega(\varepsilon^{-2})$ for $1 < p \leq 2$, and $\Omega(\varepsilon^{-2/3})$ for $p = 1$. This paper also has lower bounds for smooth convex functions.

**Submodular Function Minimization.** We now expand on the history of SFM upper and lower bounds for parallel and sequential algorithms touched upon earlier. Since the seminal work of Edmonds in 1970 [Edm70], there has been extensive work [GLS81, GLS88, Sch00, IFF01, Iwa03, FI00, Vyg03, Orl09, IO09, LSW15, CLSW17, ALS20, DVZ21, Jia21, Jia22] on developing query-efficient algorithms for SFM. [GLS81, GLS88] gave the first polynomial time algorithms using the ellipsoid method. The state-of-the-art SFM algorithms include a $\widetilde{O}(n^2)$-query exponential time algorithm due to [Jia22], $\widetilde{O}(n^3)$-query polynomial time algorithms due to [Jia21, JLSZ23]; a $\widetilde{O}(n^2 \log M)$-query polynomial time algorithm due to [LSW15], and a $\widetilde{O}(nM^2)$-query polynomial time algorithm due to [ALS20]. Despite these algorithmic improvements, limited progress has been made on lower bounding the query complexity of SFM and the best known lower bound has been $\Omega(n)$ for decades [Har08, CLSW17, GPRW20]. Very recently, [CGJS22] proved an $\Omega(n \log n)$-lower bound for deterministic SFM algorithms.

All the algorithms above are highly sequential and proceed in at least $n$ rounds. The question of parallel complexity for SFM was first studied in [BS20] where an $\Omega(\log n/ \log \log n)$-lower bound on the number of rounds required by any query-efficient SFM algorithm was given. The range $M$ in their construction is $M = n^{\Theta(n)}$. Subsequently, [CCK21] proved a $\widetilde{\Omega}(n^{1/3})$ lower bound on the round-complexity and the range is $M = n$ for their functions. Recently, [CGJS22] described a $\widetilde{\Omega}(n)$ lower bound with functions of range $M = n^{\Theta(n)}$.

**Cutting plane methods.** Cutting plane methods are a class of optimization methods that minimize a convex function by iteratively refining a convex set containing the minimizer. Since the center of gravity method was developed independently in [Lev65, New65], there have been many developments of faster cutting plane methods over the last six decades [Sho77, YN76, Kha80, KTE88, NN89, Vai89, BV04, LSW15], with the state-of-the-art due to [JLSW20].

## 2 Minimizing $\ell_\infty$-Lipschitz Functions and Submodular Functions

In this section we provide a new, improved parallel algorithm for convex optimization in $\ell_\infty$ and show how to use these algorithms to obtain an improved parallel algorithm for SFM. In much of this section, we consider the following optimization problem which we term *stochastic convex optimization in $\ell_\infty$*. As we discussed, this problem generalizes parallel convex optimization in $\ell_\infty$. The problem is more general in terms of the norms it considers and how it allows for stochastic gradients; we consider it as it could be useful more broadly and as it perhaps more tightly captures the performance of our optimization algorithm.

**Definition 2.1** (Stochastic Convex Optimization in $\ell_\infty$). *In the stochastic convex optimization in $\ell_\infty$ problem we have a (stochastic) subgradient oracle $g : \mathbb{R}^n \to \mathbb{R}^n$ such that $\mathbb{E}g(x) \in \partial f(x)$ and $\mathbb{E}\|g(x)\|_2^2 \le \sigma^2$ for a convex function $f : \mathbb{R}^n \to \mathbb{R}$ that is $L$-Lipschitz in $\ell_\infty$. Given the guarantee that $f$ has a minimizer $x_* \in \mathbb{R}^n$ with $\|x_*\|_2 \le R$ our goal is to compute an $\varepsilon$-approximate minimizer of $f$, i.e., $x \in \mathbb{R}^n$ with $f(x) \le f(x_*) + \varepsilon$.*

Note that in Definition 2.1, $\ell_\infty$ appears only to determine the norm in which $f$ is Lipschitz. However, the bound on $x_*$ in $\ell_2$ that can be easily converted to one in terms of $\ell_\infty$ by using that $\|x_*\|_2 \le \sqrt{n} \|x_*\|_\infty$. Furthermore, a convex function $f : \mathbb{R}^n \to \mathbb{R}$ is $L$-Lipschitz in $\ell_\infty$ if and only if $\|g\|_1 \le L$ for all $g \in \partial(x)$ for $x \in \mathbb{R}^n$. Since $\|g\|_2 \le \|g\|_1$ we see that this stochastic convex optimization problem subsumes the (non-stochastic) problem of computing an $\varepsilon$-approximate minimizer to a convex function that is $L$-Lipschitz in $\ell_\infty$ given a (deterministic) subgradient oracle. We define this more general problem as, interestingly, our algorithm tolerates this weaker stochastic oracle without any loss (as we discussed).

Our main result regarding stochastic convex optimization in $\ell_\infty$ is given in the following theorem.

**Theorem 2.2** (Stochastic Convex Optimization in $\ell_\infty$). *There is an algorithm that solves the stochastic convex optimization problem (Definition 2.1) in $\ell_\infty$ (Definition 2.1) in $\widetilde{O}((LR/\varepsilon)^{2/3})$ rounds and $\widetilde{O}((\sigma R/\varepsilon)^2)$ queries.*

Due to the aforementioned connection between $\ell_\infty$-Lipschitz continuity and bounds on the subgradient, and the fact that $\|x_*\|_\infty \le 1$ implies $\|x_*\|_2 \le \sqrt{n}$, Theorem 2.2 immediately yields a $\widetilde{O}(n^{1/3}\varepsilon^{-2/3})$-round, $\widetilde{O}(n\varepsilon^{-2})$-query algorithm for the problem of minimizing a convex function that is 1-Lipschitz in $\ell_\infty$ and minimized at a point with $\ell_\infty$-norm at most 1. As discussed in the introduction, the parallel complexity of this algorithm is near-optimal for constant $\varepsilon$ [Nem94].

**Theorem 1.2** (Parallel Convex Optimization in $\ell_\infty$). *There is an algorithm that when given a subgradient oracle for convex $f : \mathbb{R}^n \to \mathbb{R}$ that is 1-Lipschitz in $\ell_\infty$ and has a minimizer $x_*$ with $\|x_*\|_\infty \le 1$ computes an $\varepsilon$-approximate minimizer of $f$ in $\widetilde{O}(n^{1/3}\varepsilon^{-2/3})$ rounds and $\widetilde{O}(n\varepsilon^{-2})$ queries.*

In Section 2.1, we show how to use Theorem 2.2 to obtain our results for SFM. We then present the ingredients in the proof of Theorem 2.2 (which we defer to Appendix A) in Section 2.2. As part of our reduction from SFM to Stochastic Convex Optimization in $\ell_\infty$ in Theorem 2.2, we provide a general tool for reducing constrained to unconstrained minimization (Lemma 2.3); we use this lemma to facilitate our results in both sections.

## 2.1 From Unconstrained Convex Optimization in $\ell_\infty$ to SFM

Here we show how to use Theorem 1.2 to prove the following theorem regarding SFM.

**Theorem 1.3** (Sublinear Parallel SFM). *There is an algorithm that, when given an evaluation oracle for submodular $f : 2^{[n]} \to \mathbb{Z}$ with $f(\emptyset) = 0$ and $|f(S)| \le M$ for all $S \subseteq [n]$, finds a minimizer of $f$ in $\widetilde{O}(n^{1/3}M^{2/3})$ rounds and $\widetilde{O}(n^2 M^2)$ queries.*

A key ingredient of our proof is the following general, simple technical tool which allows one to reduce constrained Lipschitz optimization over a ball in any norm to unconstrained minimization with only a very mild increase in parameters.

**Lemma 2.3.** *Let $f : \mathbb{R}^n \to \mathbb{R}$ be convex and $L$-Lipschitz with respect to norm $\|\cdot\| : \mathbb{R}^n \to \mathbb{R}$. For any $c, x \in \mathbb{R}^n$ and $r \in \mathbb{R}$ let*

$$f_{\text{reg}}^{c,r}(x) := f(x) + 2L \cdot \max\{0, \|x - c\| - r\}. \tag{1}$$

*Then $f_{\text{reg}}^{c,r}(x)$ is convex and $3L$-Lipschitz with respect to $\|\cdot\|$. Additionally, for any $y \in \mathbb{R}^n$ for which $\|y - c\| \ge r$, define $y^{c,r} := c + \frac{r}{\|y-c\|}(y - c)$. Then,*

$$\|y^{c,r} - c\| = r \text{ and } f_{\text{reg}}^{c,r}(y^{c,r}) = f(y^{c,r}) \le f_{\text{reg}}^{c,r}(y) - L(\|y - c\| - r). \tag{2}$$

*Consequently, $f_{\text{reg}}^{c,r}(x)$ has an unconstrained minimizer $x_*^{c,r}$ and all such minimizers satisfy*

$$\|x_*^{c,r} - c\| \le r \text{ and } f_{\text{reg}}^{c,r}(x_*^{c,r}) = f(x_*^{c,r}) = \min_{x \in \mathbb{R}^n \mid \|x-c\| \le r} f(x). \tag{3}$$

Lemma 2.3 implies that minimizing $f$ subject to a distance constraint $\|x - c\| \leq r$ reduces to unconstrained minimization of $f_{\text{reg}}^{c,r}$. More formally, to compute an $\epsilon$-optimal solution to the constrained minimization problem, $\min_{x \in \mathbb{R}^n : \|x-c\| \leq r} f(x)$, it suffices to instead compute an $\varepsilon$-optimal solution, $x_\varepsilon$, to the unconstrained minimization problem $\min_x f_{\text{reg}}^{c,r}(x)$, and then output that point $x_\varepsilon$ if $\|x_\varepsilon - c\| \leq r$ and output $x_\varepsilon^{r,c}$-otherwise. From (2), we get $f(x_\varepsilon^{r,c}) \leq f_{\text{reg}}^{c,r}(x_\varepsilon) \leq \text{opt} + \varepsilon$, where $\text{opt} := \min_x f_{\text{reg}}^{c,r}(x) = \min_{x \in \mathbb{R}^n | \|x-c\| \leq r} f(x)$ and the last equality follows from (3).

We remark that the $2L$ in the definition of $f_{\text{reg}}^{c,r}$ can be changed to $L + \delta$ for any $\delta \geq 0$ with the only effect of turning the $L$ in (2) to $\delta$ and causing (3) to only hold for some minimizer (rather than all) if $\delta = 0$. The proof of Lemma 2.3 is deferred to Appendix A.

Next, we obtain Theorem 1.3 by applying Theorem 2.2 to the Lovász extension of the submodular function $f$ extended to an unconstrained minimization problem by Lemma 2.3.

Given a submodular function $f$ defined over subsets of an $n$ element universe, the Lovász extension $f_{\text{Lov}} : \mathbb{R}^n \to \mathbb{R}$ is defined as follows: $f_{\text{Lov}}(x) := \sum_{i \in [n]} x_{\pi_x(i)}(f(S_{\pi_x,i}) - f(S_{\pi_x,i-1}))$, where $\pi_x : [n] \to [n]$ is the permutation such that $x_{\pi_x(1)} \geq x_{\pi_x(2)} \geq \cdots \geq x_{\pi_x(n)}$ (ties broken in an arbitrary but consistent manner), and $S_{\pi_x,j}$ is the subset $\{\pi_x(1), \ldots, \pi_x(j)\}$.

Next we give standard properties of the Lovász extension and use them to prove Theorem 1.3.

**Fact 2.4** (e.g., [GLS88, JB11]). *The following are true about the Lovász extension $f_{\text{Lov}}$:*

1. *$f_{\text{Lov}}$ is convex with $\min_{x \in [0,1]^n} f_{\text{Lov}}(x) = \min_{S \subseteq V} f(S)$. Indeed, given any $x \in [0,1]^n$, in $n$ queries one can find a subset $S$ with $f(S) \leq f_{\text{Lov}}(x)$.*
2. *Given any $x \in \mathbb{R}^n$ and corresponding permutation $\pi_x$, the vector $g \in \mathbb{R}^n$ where $g(x)_{(\pi(i))} := f(S_{\pi_x,i}) - f(S_{\pi_x,i-1})$ is a subgradient of $f_{\text{Lov}}$ at $x$. Furthermore, $g(x)$ can be computed in 1 round of $n$ queries to an evaluation oracle for $f$.*
3. *If $f$ has range in $[-M, +M]$, then the $\ell_1$-norm of the subgradient is bounded, in particular, $\|g(x)\|_1 \leq 3M$. Equivalently, $f_{\text{Lov}}$ is $3M$-Lipschitz with respect to the $\ell_\infty$-norm.*

As discussed in Section 1.1, to prove Theorem 1.3, it suffices to prove the following more general result regarding approximately minimizing a submodular function.

**Theorem 2.5** ($\epsilon$-approximate minimizer for SFM). *There is an algorithm that, when given an evaluation oracle for submodular $f : 2^{[n]} \to \mathbb{R}$ with minimizer $x_*$, $f(\emptyset) = 0$ and $|f(S)| \leq M, \forall S \subseteq [n]$, finds a set $S$ with $f(S) \leq f(x_*) + \epsilon M$ in $\widetilde{O}(n^{1/3}/\epsilon^{2/3})$ rounds and a total of $\widetilde{O}(n^2/\epsilon^2)$ queries.*

*Proof of Theorem 2.5.* By Fact 2.4, SFM reduces to minimizing $f_{\text{Lov}}$ over $x \in [0,1]^n$ which is the same set $\{x \in \mathbb{R}^n : \|x - c\|_\infty \leq 0.5\}$ where $c$ is the $n$-dimensional vector with all entries $0.5$. By Lemma 2.3, we can do so by applying Theorem 2.2 to the regularized version $f_{\text{reg}}^{c,0.5}$ of $f_{\text{Lov}}$ with respect to the $\ell_\infty$-norm. This regularized function is guaranteed to have a minimizer $x_*$ with $\|x_*\|_2 \leq \sqrt{n}$ which also minimizes $f_{\text{Lov}}$ in $[0,1]^n$. The subgradient of this regularized function at any point can be computed from the subgradient of $f_{\text{Lov}}$ at the same point which takes $n$ evaluation oracle queries to the submodular function. Hence, we obtain an $\varepsilon M$-approximate minimizer in $\widetilde{O}(n^{1/3}/\varepsilon^{2/3})$ rounds and with a total of $\widetilde{O}(n^2/\varepsilon^2)$ queries to the evaluation oracle of $f$. $\qquad \square$

Now, we are ready to prove Theorem 1.3.

*Proof of Theorem 1.3.* Set $\epsilon = \frac{1}{2M}$ and apply Theorem 2.5 to obtain, in $\widetilde{O}(n^{1/3}M^{2/3})$ rounds and with a total of $\widetilde{O}(n^2 M^2)$ queries to the evaluation oracle of $f$, a $x \in [0,1]^n$ with $f_{\text{Lov}}(x) \leq \min_{z \in [0,1]^n} f_{\text{Lov}}(z) + \frac{1}{2}$. Then by property 1 in Fact 2.4, one can get a subset $A \subseteq V$ with $f(A) \leq \min_{S \subseteq V} f(S) + \frac{1}{2}$. As $f$ is assumed to be integer valued, $A$ must be the minimizer of the submodular function. $\qquad \square$

## 2.2 Parallel Stochastic Convex Optimization in $\ell_\infty$

Here we present the key steps in proving Theorem 2.2 regarding our new parallel results for the stochastic convex optimization problem in $\ell_\infty$ (Definition 2.1). Throughout this subsection, in our exposition, lemma statements, and proofs we assume that we are in the setting of Definition 2.1.

To prove Theorem 2.2 we apply the approach of [CJJ+23] with two modifications. First, we consider the convolution of $f$ with a centered Gaussian density function with covariance $\rho^2 \mathbf{I}_n$. However we show that in our setting, it is possible to use a larger value of $\rho$ without perturbing the function value too much, due to the $\ell_\infty$ geometry. Unfortunately, the minimizer of the convolved function may move outside the box of radius $R$. Thus, the second modification we make is working with a regularized function, $f_{\text{reg}}$, which is pointwise close to $f$, still $L$-Lipschitz in the $\ell_\infty$ norm, and keeps the minimizer in the ball or radius $R$ even after applying the convolution with a Gaussian.

In the remainder of this subsection we first present the ingredients going into the proof of Theorem 2.2, and then give a brief explanation for how they fit into the proof of Theorem 2.2, deferring the complete proof to the appendix. We start with our bound on function perturbation after adding a Gaussian with covariance $\rho^2 \mathbf{I}_n$ to an $\ell_\infty$ Lipschitz function (Lemma 2.6). We then introduce the concept of a ball optimization oracle, along with a result on how to implement it in low depth for the special case of a function that is the result of Gaussian convolution (Proposition 2.8). Lastly, we present the result which allows us to use a ball optimization oracle black-box to obtain the desired depth (Proposition 2.9).

Now, we are ready to present the lemma which allows us to obtain a better dependence of depth on the dimension $n$, compared to the naive $n^{2/3}$ obtained by directly applying the $\ell_2$-Lipschitz optimization result. We start with the definition of Gaussian convolution.

**Gaussian Convolution.** Let $\gamma_\rho := (2\pi\rho)^{-n/2} \exp(-\frac{1}{2\rho^2} \|x\|_2^2)$ be the probability density function of $\mathcal{N}(0, \rho^2 \mathbf{I}_n)$. Given a function $f : \mathbb{R}^n \to \mathbb{R}$, we define its convolution with a Gaussian of covariance $\rho^2 \mathbf{I}_n$ by $\widehat{f}_\rho := f * \gamma_\rho$, i.e.

$$\widehat{f}_\rho(x) := \mathbb{E}_{y \sim \mathcal{N}(0,\rho^2 \mathbf{I}_n)}[f(x+y)] = \int_{\mathbb{R}^n} f(x-y)\gamma_\rho(y)\mathrm{d}y. \tag{4}$$

Next, we present a lemma which allows us to obtain a better dependence on the dimension $n$ in depth, compared to the naive $n^{2/3}$ obtained by directly applying the $\ell_2$-Lipschitz optimization result. This lemma shows that we can perform more Gaussian smoothing (as compared to the $\ell_2$-setting) without perturbing the function too much (as mentioned in Section 1.1).

**Lemma 2.6** (Gaussian Convolution Distortion Bound for $\ell_\infty$). *Let $f : \mathbb{R}^n \to \mathbb{R}$ be $L$-Lipschitz with respect to the $\ell_\infty$-norm. Then for any point $x \in \mathbb{R}^n$, we have $|\widehat{f}_\rho(x) - f(x)| \leq L\rho \cdot \sqrt{2 \log n}$.*

*Proof.* Note that $|\widehat{f}_\rho(x) - f(x)| \leq \int_{\mathbb{R}^n} |f(z) - f(x)|\gamma_\rho(x-z)\mathrm{d}z \leq L \int_{\mathbb{R}^n} \|x-z\|_\infty \gamma_\rho(x-z)\mathrm{d}z$ where the first inequality follows from the definition of $\widehat{f}_\rho$ and the second follows as $f$ is $L$-Lipschitz in $\ell_\infty$-norm. The RHS is simply the expected $\ell_\infty$-norm of a zero-mean random Gaussian vector with covariance $\rho^2 \mathbf{I}_n$, and this is $\Theta(\rho\sqrt{\log n})$ (e.g., [Ver18]). More precisely, we get

$$|\widehat{f}_\rho(x) - f(x)| \leq L \cdot \mathbb{E}_{y \sim \mathcal{N}(0,\rho^2 \mathbf{I}_n)} \|y\|_\infty \leq L\rho \cdot \sqrt{2 \log n}. \qquad \square$$

As mentioned in Section 1.1, by contrast, convolving a function $f$ that is $L$-Lipschitz in $\ell_2$ with a Gaussian of covariance $\rho^2 \mathbf{I}_n$ could change the function value by $O(\rho\sqrt{n})$. Hence, the $\ell_\infty$ geometry allows us to add a larger amount of Gaussian smoothing without changing the function value by more than $\varepsilon$, which in turn allows for better rates.

**Ball Optimization.** A subroutine that we use for minimizing $\widehat{f}_\rho$ is called a *ball optimization oracle*. As suggested by [CJJ+20], the concept of ball optimization oracle is related to the notion of trust regions, explored in several papers, such as [CGT00]. The particular ball optimization procedure we employ takes a function $F : \mathbb{R}^n \to \mathbb{R}$ and a point $\bar{x} \in \mathbb{R}^n$, which is an approximate solution to $F$ in a small ball of $\bar{x}$. More formally, we work with the following definition:

**Definition 2.7** (Ball Optimization Oracle [CJJ+23]). *Let $F : \mathbb{R}^n \to \mathbb{R}$ be a convex function. $\mathcal{O}_{\text{bo}}$ is an $(\phi, \lambda, r)$-ball optimization oracle for $F$ if given any $\bar{x} \in \mathbb{R}^n$, it returns an $x \in \mathbb{R}^n$ with the property*

$$\mathbb{E}\left[F(x) + \frac{\lambda}{2} \cdot \|x - \bar{x}\|_2^2\right] \leq F(x_{\text{loc}}^\star) + \frac{\lambda}{2} \|x_{\text{loc}}^\star - \bar{x}\|_2^2 + \phi,$$

*where $x_{\text{loc}}^\star = \arg\min_{x \in B_{\bar{x}}(r)}(F(x) + \frac{\lambda}{2} \|x - \bar{x}\|_2^2)$.*

From [CJJ+20] it is known is that for any Lipschitz convex function $f$, any stochastic subgradient oracle $g$ as above, and any $\rho$, if we set $r = \rho$, then efficient ball-optimization oracles exist. More formally, we use the following proposition from [CJJ+23] which is in turn inspired from [ACJ+21].

**Proposition 2.8** (Proposition 3, [CJJ+23])**.** *Let $f : \mathbb{R}^n \to \mathbb{R}$ be convex and $g : \mathbb{R}^n \to \mathbb{R}^n$ be a stochastic subgradient oracle satisfying $\mathbb{E}[g(x)] \in \partial f(x)$ and $\mathbb{E}\|g(x)\|_2^2 \leq \sigma^2$ for all $x \in \mathbb{R}^n$. Let $\widehat{f}_\rho := f * \gamma_\rho$, i.e.,*

$$\widehat{f}_\rho(x) := \mathbb{E}_{y \sim \mathcal{N}(0, \rho^2 \mathbf{I}_n)}[f(x - y)] = \int_{\mathbb{R}^n} f(x - y)\gamma_\rho(y)\mathrm{d}y. \tag{5}$$

*Then there is a $(\phi, \lambda, \rho)$-ball optimization oracle for $\widehat{f}_\rho$ which makes $O(\frac{\sigma^2}{\phi\lambda})$ total queries to $g$ in a constant number of rounds.*

**Highly Parallel Optimization.** As shown in [CJJ+23], the ball optimization oracle above can be used for highly-parallel optimization as follows.

**Proposition 2.9** (Proposition 2 in [CJJ+23])**.** *Fix a function $F : \mathbb{R}^n \to \mathbb{R}$ which is $L$-Lipschitz with respect to the $\ell_2$-norm and convex. Suppose $R \geq 0$ is a parameter such that $x^\star \in \arg\min_x F(x)$ satisfies $\|x^\star\|_2 \leq R$. Let $r \in (0, R]$ and $\epsilon_{\mathsf{opt}} \in (0, LR]$ be two parameters. Define the following quantities*

$$\kappa := \frac{LR}{\epsilon_{\mathsf{opt}}}, \quad K := \left(\frac{R}{r}\right)^{\frac{2}{3}}, \quad and \quad \lambda_* := \frac{\epsilon_{\mathsf{opt}} K^2}{R^2} \log^2 \kappa. \tag{6}$$

*Then, there exists a universal constant $C > 0$ and an algorithm $\mathsf{BallAccel}$ which runs in $CK \log \kappa$ iterations and produces a point $x$ such that $\mathbb{E}F(x) \leq F(x^\star) + \epsilon_{\mathsf{opt}}$. Moreover,*

1. *Each iteration makes at most $C \log^2(\frac{R\kappa}{r})$ calls to $(\frac{\lambda r^2}{C}, \lambda, r)$-ball optimization oracle with values of $\lambda \in [\frac{\lambda_*}{C}, \frac{CL}{\epsilon_{\mathsf{opt}}}]$.*

2. *For each $j \in [\lceil \log_2 K + C \rceil]$, at most $C^2 \cdot 2^{-j} K \log(\frac{R\kappa}{r})$ iterations query a $(\frac{\lambda r^2}{C2^j} \cdot \log^{-2}(\frac{R\kappa}{r}), \lambda, r)$-ball optimization oracle for some $\lambda \in [\frac{\lambda_*}{C}, \frac{CL}{\epsilon_{\mathsf{opt}}}]$.*

While the proof of Theorem 2.2 is deferred to Appendix A, we provide some intuition for how to use the stated components to obtain the result. A natural approach would be to apply Proposition 2.9 to $F := \widehat{f}_\rho$. However, to ensure that the minimizer of $F$ has $\ell_2$-norm at most $R$, we will instead work with $F := \widehat{f}^{c,R}_{\mathsf{reg}_\rho}$ as defined in Lemma 2.3. The idea is to invoke Lemma 2.3 and apply $\mathsf{BallAccel}$ on the function $F := \widehat{f}^{c,R}_{\mathsf{reg}_\rho}$ with respect to $\|\cdot\|_2$, $c$ being the origin, and $\rho := \frac{\epsilon_{\mathsf{opt}}}{L\sqrt{2\log n}}$. With this choice of $\rho$, by Lemma 2.6, we know $|F(x) - f^{c,R}_{\mathsf{reg}}(x)| \leq \epsilon_{\mathsf{opt}}$ everywhere. Noting that $F$ has a minimizer $x_*$ with $\|x_*\|_2 \leq 3R$, this enables us to apply Proposition 2.9 to $F$, obtaining the stated bounds on the number of rounds and query complexity.

# 3    2-Round $O(n^{M+1})$-Query Algorithm for SFM

Here we present our 2-round, $O(n^{M+1})$-query algorithm for SFM. The algorithm AugmentingSets, given in Algorithm 1, iterates over every $M$-sparse $S \subseteq [n]$ (i.e. $|S| \leq M$) (denoted $\mathcal{F}$). For every such $S$ the algorithm then builds the augmented set $A(S)$, consisted of the union of $S$ and all elements $i$ that have non-positive marginal with respect to $S$, i.e., $f(S \cup \{i\}) \leq f(S)$. The algorithm then outputs the set $A(S)$ that has the smallest value.

As we show below, computing all the $A(S)$ for $M$-sparse sets $S$ can be done in 1 round and $O(n^{M+1})$-queries, and then computing an element of $A(S)$ with the smallest value can be done in another round and $O(n^{M+1})$-queries. The correctness of the algorithm is guaranteed by the fact that $A(S)$, for some $|S| \leq M$, is the maximal minimizer of $f$, and therefore the algorithm outputs a set with the optimum value[6].

Our main result of this section is the following theorem.

---

[6]Note, however, that the algorithm doesn't necessarily output the maximal minimizer itself.

**Algorithm 1:** Augmenting Sets Algorithm

---

**Input:** Submodular function $f : 2^{[n]} \to \mathbb{Z}$ and $M \in \mathbb{Z}_{>0}$ such that $|f(S)| \le M$ for all $S \subseteq [n]$ and $f(\emptyset) = 0$

**Output:** $S_{\text{out}}$, a minimizer of function $f$

1 **Function** AugmentingSets($f, M$):
2     $\mathcal{F} \leftarrow \{S \subset [n] \mid |S| \le M\}$
3     **for** $S \in \mathcal{F}$ **do**
4        $A(S) \leftarrow S \cup \{i \notin S \mid f(S \cup \{i\}) \le f(S)\}$    // Compute the augmentation of $S$
5     **end**
6     **return** $S_{\text{out}} \in \operatorname{argmin}_{S \in \mathcal{F}} f(A(S))$

---

**Theorem 1.4** (Two-round Parallel SFM). *There is an algorithm (Algorithm 1) that when given an evaluation oracle for submodular $f : 2^{[n]} \to \mathbb{Z}$ with $f(\emptyset) = 0$ and $|f(S)| \le M$ for all $S \subseteq [n]$ finds a minimizer of $f$ in 2 rounds and $O(n^{M+1})$ queries.*

*Proof.* First, we bound the parallel and query complexity of the algorithm. Line 2 to Line 5 can be implemented in 1 round as they simply evaluate $f$ on all subsets of $[n]$ of size $\le M + 1$. Line 6 can be implemented in 1 round by evaluating $f(A(S))$ for each $S \in \mathcal{F}$ in parallel. Consequently, the algorithm is implementable in 2 rounds. To bound the query complexity, note that $|\mathcal{F}| \le \sum_{k=0}^{M} \binom{n}{k} = O(n^M)$ and the algorithm only makes $O(n)$ queries for each $S \in \mathcal{F}$ ($O(n)$ in Line 4 and 1 in Line 6). Consequently, the algorithm makes $O(n^{M+1})$ total queries.

It only remains to show that the algorithm outputs a minimizer of $f$. We prove this by showing that for some $S \in \mathcal{F}$, its augmented set $A(S)$ is the maximal minimizer of $f$, i.e., the union of all minimizers, which is a minimizer itself by submodularity. This suffices as the algorithm outputs the $A(S)$ for $S \in \mathcal{F}$ of smallest value. Let $S_*$ be the maximal minimizer of $f$. We build a subset $T \subseteq S_*$ of size $|T| \le M$, which we call *anchor*, as follows. Start with $T = \emptyset$ and an arbitrary ordering of elements of $S_*$. For each element $i \in S_*$ in this order, we add it to the current $T$ if $f(T \cup \{i\}) > f(T)$. Since $f$ only takes integer values, this means that whenever we add an element $i$ to $T$, the value of $f(T)$ goes up by at least 1. At the end of the process we have $f(T) \ge |T|$. Since $f(T) \le M$, it follows that $|T| \le M$ and therefore $T \in \mathcal{F}$.

We now claim that $A(T) = S_*$. For any element $i \in S_* \setminus T$, we didn't add $i$ to $T$ because $f(T_i \cup \{i\}) - f(T_i) \le 0$, where $T_i \subseteq T$ is the value of $T$ when element $i$ is visited in the procedure above. By submodularity, $f(T \cup \{i\}) - f(T) \le f(T_i \cup \{i\}) - f(T_i) \le 0$. This implies that $S_* \subseteq A(T)$. Also note that for any $j \notin S_*$, we have $f(S_* \cup \{j\}) > f(S_*)$ by the *maximality* of $S_*$. It again follows from submodularity and $T \subseteq S_*$ that $f(T \cup \{j\}) > f(T)$, which implies $j \notin A(T)$. This proves $A(T) = S^*$ and completes the proof of the theorem. $\qquad\square$

## 4 Conclusion

In this paper we designed two new parallel algorithms for minimizing submodular functions $f : 2^{[n]} \to \mathbb{Z} \cap [-M, +M]$ with round complexities $\widetilde{O}(n^{1/3} M^{2/3})$ and 2, and query complexities $\widetilde{O}(n^2 M^2)$ and $O(n^{M+1})$, respectively. These $M$-dependent sublinear dependence on $n$ in the round complexities stand in contrast to the $\widetilde{\Omega}(n)$-lower bound on the number of rounds required for SFM when $M = n^{\Theta(n)}$. On the way to the first result, we obtain a new efficient parallel algorithm for $\varepsilon$-approximate minimization of $\ell_\infty$-Lipschitz convex functions over $[0, 1]^n$ with round-complexity $\widetilde{O}(n^{1/3} \varepsilon^{-2/3})$. Given results of [Nem94, DG19] the dependence on $n$ is optimal for constant $\varepsilon$.

Two related open questions are whether one can obtain $o(n)$-round SFM algorithms with polylogarithmic dependence on $M$, and whether one can obtain algorithms for $\varepsilon$-approximate minimization of $\ell_\infty$-Lipschitz convex functions over $[0, 1]^n$ in $\widetilde{O}(n^{1/3} \operatorname{poly} \log(1/\varepsilon))$-rounds, or can one prove lower bounds ruling them out. Another related open question is whether one can perform SFM in $O(\operatorname{poly}(M))$-rounds with query complexity $\operatorname{poly}(n)$.

## Acknowledgements

We thank the anonymous reviewers of NeurIPS 2023 for helpful comments. Part of this work was done while Deeparnab Chakrabarty, Haotian Jiang, and Aaron Sidford were attending the Discrete Optimization trimester program at the Hausdorff Research Institute for Mathematics.

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

# A Supplement

In this supplement, we provide the complete proofs of Lemma 2.3 and Theorem 2.2.

*Proof of Lemma 2.3.* First we show that $f_{\text{reg}}^{c,r}(x)$ is convex and $3L$-Lipschitz. Let $h(x) := 2L(\|x - c\| - r)$. By triangle inequality and homogenity of norms, $2L(\|x - c\| - r)$ is convex and $2L$-Lipschitz (with respect to $\|\cdot\|$). Further, it is straightforward to check that the maximum of two convex, $L$-Lipschitz functions is convex and $L$-Lipschitz and that the sum of a $L$-Lipschitz and $2L$-Lipschitz function is convex and $3L$-Lipschitz. Since $f_{\text{reg}}^{c,r}(x) = f(x) + \max\{0, h(x)\}$ and constant functions are convex and $0$-Lipschitz the result follows.

Next, consider $y \in \mathbb{R}^n$ with $\|y - c\| \geq r$. Since $\|\cdot\|$ is a norm we see that $\|y^{c,r} - c\| = r$ and therefore $f_{\text{reg}}^{c,r}(y^{c,r}) = f(y^{c,r})$. Furthermore, $\|y^{c,r} - y\| = \|y - c\| - r$ and so,

$$f_{\text{reg}}^{c,r}(y) = f(y) + 2L(\|y - c\| - r) \geq f(y^{c,r}) - L\|y^{c,r} - y\| + 2L(\|y - c\| - r)$$
$$= f(y^{c,r}) + L(\|y - c\| - r).$$

where the inequality follows since $f$ is $L$-Lipschitz with respect to $\|\cdot\|$. This yields the desired result as it implies that values of $f_{\text{reg}}^{c,r}$ are all larger than those of points where $\|x - c\| \leq r$ for which $f_{\text{reg}}^{c,r}(x) = f(x)$. Further, since $\{x \in \mathbb{R}^n | \|x - c\| \leq r\}$ is closed and $f$ is Lipschitz, ther exists a minimizer of $\min_{x \in \mathbb{R}^n | \|x - c\| \leq r} f(x)$ and the result follows. $\qquad\square$

*Proof of Theorem 2.2.* We invoke Lemma 2.3 and apply BallAccel on the function $F := \widehat{f}_{\text{reg}_\rho}^{c,R}$ as defined in Lemma 2.3 with respect to $\|\cdot\|_2$, $c$ being the origin, and $\rho := \frac{\epsilon_{\text{opt}}}{L\sqrt{2\log n}}$. With this choice of $\rho$, by Lemma 2.6, we know $|F(x) - f_{\text{reg}}^{c,R}(x)| \leq \epsilon_{\text{opt}}$ everywhere. For brevity, we remove the superscript $(c, R)$ from $f_{\text{reg}}$ for the remainder of the proof.

We now claim that $F$ has a minimizer $x_*$ with $\|x_*\|_2 \leq 3R$. To see this, let $y \in \mathbb{R}^n$ be a point with $\|y\|_2 > 3R$ and consider $\tilde{y} = \frac{y}{\|y\|_2} \cdot R$. Therefore,

$$F(y) \geq f_{\text{reg}}(y) - \epsilon_{\text{opt}} = f(y) + 2L \cdot (\|y\|_2 - R) - \epsilon_{\text{opt}} \geq f_{\text{reg}}(\tilde{y}) + L \cdot (\|y\|_2 - R) - \epsilon_{\text{opt}}$$

where the first inequality follows from pointwise approximation of $F$ and $f_{\text{reg}}$, and the second follows since $f$ is also $L$-Lipschitz in the $\|\cdot\|_2$ norm, and so $f_{\text{reg}}(\tilde{y}) = f(\tilde{y}) \leq f(y) + L \cdot (\|y\|_2 - R)$. Again using the pointwise approximation of $F$ and $f_{\text{reg}}$ we get $F(y) \geq F(\tilde{y}) - 2\epsilon_{\text{opt}} + L \cdot (\|y\|_2 - R)$, and if $\|y\|_2 > 3R$ using that $\epsilon_{\text{opt}} \in (0, LR]$ we get a contradiction to minimality of $F(y)$.

Note that the stochastic subgradient $g'$ of $f_{\text{reg}}$ is given by $g' = g + 2L \cdot v$ for $v \in \partial h(x)$ where $h(x) = \max(0, \|x\|_2 - R)$. Note that $\|v\|_2^2 \leq 1$, so the stochastic gradient $g'$ satisfies $\mathbb{E}\|g'(x)\|_2^2 \leq 2\sigma^2 + 8L^2$. It follows that by setting $r = \rho$, Proposition 2.8 implies that for any $\phi, \lambda > 0$, there exists a $(\phi, \lambda, \rho)$-ball optimization oracle for $F$ which makes $O(\frac{\sigma^2}{\phi\lambda})$ total queries to $g$ in $O(1)$ parallel rounds (as $\sigma^2 \geq L^2$ by definition of $L$-Lipschitzness).

Next we apply ball acceleration in Proposition 2.9 to $F$. We have already argued above that the minimizer $x_*$ of $F$ satisfies $\|x_*\|_2 \leq 3R$. We set the parameter $r = \rho = \frac{\epsilon_{\text{opt}}}{L\sqrt{2\log n}}$ and the $R$ and $L$ multiplied by factor 3. Using these, the parameters of (6) in Proposition 2.9 become

$$\kappa = \frac{LR}{\epsilon_{\text{opt}}}, \quad K = \left(\frac{LR\sqrt{2\log n}}{\epsilon_{\text{opt}}}\right)^{2/3}, \quad \text{and } \lambda_\star = \frac{\epsilon_{\text{opt}}K^2}{R^2} \cdot \log^2 \kappa.$$

Using Proposition 2.9 and Proposition 2.8, we get that the number of rounds of queries is $(CK \log \kappa) \cdot (C \log^2(R\kappa/\rho))$ which is $\widetilde{O}((LR/\epsilon_{\text{opt}})^{2/3})$.

Next, we bound the query complexity. Adding up the queries made by the ball-optimization oracle calls made in all iterations per part 1 of Proposition 2.9 is

$$(CK \log \kappa) \cdot \left(C \log^2(R\kappa/\rho)\right) \cdot \frac{\sigma^2}{\lambda_\star^2 \rho^2} = \widetilde{O}\left(K\sigma^2 \lambda_\star^{-2} \rho^{-2}\right).$$

Adding up the queries made by the ball-optimization oracle calls made in all iterations per part 2 of Proposition 2.9 is

$$\sum_{j\in[\lceil\log_2 K+C\rceil]}\left(C^2 2^{-j}K\log(R\kappa/\rho)\right)\cdot\frac{2^j C\sigma^2\log^2(R\kappa/\rho)}{\lambda_\star^2\rho^2}\quad\text{which is also }\widetilde{O}\left(K\sigma^2\lambda_\star^{-2}\rho^{-2}\right).$$

Substituting the values from above and $\rho=\frac{\epsilon_{\mathsf{opt}}}{L\sqrt{2\log n}}$, we get that the total query complexity is $\widetilde{O}((\sigma R/\epsilon_{\mathsf{opt}})^2)$.

Finally note that Proposition 2.9 applied to $F$ computes an $\epsilon_{\mathsf{opt}}$-approximate minimizer which is a $2\epsilon_{\mathsf{opt}}$-approximate minimizer of $f_{\mathsf{reg}}$. By setting $\epsilon_{\mathsf{opt}}=\varepsilon/2$, Lemma 2.3 implies that we can find an $\varepsilon$-approximate minimizer $x$ of $f$ with $\|x\|_2\leq R$. This completes the proof of the theorem. $\qquad\square$

