# OpenReview forum: "Parallel Submodular Function Minimization"
_NeurIPS.cc/2023/Conference — NeurIPS 2023 spotlight_

### Official Review · Reviewer_xv3B · 2023-06-13

**Soundness:** 3 good
**Presentation:** 3 good
**Contribution:** 3 good
**Rating:** 6
**Confidence:** 4

**Summary:**

This paper considers the problem of parallel submodular function minimization, where the function is assumed to be integer valued with range $[-M, M]$. The authors propose two algorithms; one which runs in $\tilde{O}(n^{1/3} M^{2/3})$ rounds with $\tilde{O}(n^2 M^2)$ query complexity, and another which runs in 2 rounds with $n^O(M)$ query complexity. This improves over existing polynomial time submodular minimization algorithm which run in $\Omega(n)$ rounds.

**Strengths:**

The problem of minimizing submodular functions efficiently in a highly parallel manner is a natural one to consider. Several lower bounds have been derived for this problem, while upper bounds are not well investigated, except for ones resulting from sequential algorithms which run in $\Omega(n)$ rounds, and brute force search which runs in 1 round but uses $2^n$ queries. The algorithms proposed in this paper are the first upper bounds for this problem to improve over these results.
The first algorithm result also follows from improving the parallel complexity for minimizing $\ell_\infty$-Lipschitz convex functions from $\tilde{O}(n^{2/3} / \epsilon^{2/3})$ to $\tilde{O}(n^{1/3} / \epsilon^{2/3})$ rounds, which is tight in terms of dependence on n.

The results are correct and presented clearly.

**Weaknesses:**

The results of this paper are mostly based on existing work and two rather simple observations: a reduction from constrained to unconstrained optimization of L-Lipschitz functions and that convolving an $\ell_\infty$-Lipschitz function with a Gaussian changes the function less than in the case of $\ell_2$-Lipschitz functions.

The proposed algorithms are mostly of theoretical interest, I expect that the first algorithm would not be efficient in practice (it is not clear how large is the universal constant C), and the second algorithm is essentially an exhaustive search over all M-cardinality sets, which is only efficient for very small M.


**Questions:**

- Does Theorem 1.1 extends to minimizing non-integral valued submodular functions up to epsilon additive error? As far as I can tell the same proof holds. If yes, it would be good to include that.
- [ALS20] showed how to compute a sparse stochastic gradient of the lov\'asz extension of a submodular function using $O(1)$ queries. Would it be possible to reduce the query complexity of the first algorithm using a similar strategy as in  [ALS20]?

Minor issues:
- The section on related work is a bit repetitive. I suggest only mentioning additional related work not already discussed in the introduction.
- In Lemma 2.3, RHS of Eq (1), $f^{c,r}\_{reg}(y^{c,r})$ should be  $f^{c,r}\_{reg}(y)$
- In definition 1.2, $\| f(x) - f(y)|$ should be $| f(x) - f(y)|$
- In Lemma 2.5 and the discussion above it, you use d for the dimension, instead of n as in the rest of the paper.
- In Eq. (3), the term inside the expectation should be $f(x - y)$

**Limitations:**

The authors clearly state the theoretical complexity of their algorithms, along with the assumptions required for their results to hold. But they don't address the practical performance of their algorithms

---

> ### Author Rebuttal · Authors · 2023-08-09
>
> We thank the reviewer for a careful reading of our paper and writing the review.
>
> In the weakness section, the reviewer points out that our results follow from existing work and two observations namely (i) our reduction from constrained to unconstrained for Lipschitz functions and (ii) that we can convolve with a larger Gaussian for $\ell_\infty$ Lipschitz functions. We believe that our broader contribution is the connection we make between parallel SFM and $\ell_\infty$-optimization and the near optimal depth algorithm for $\ell_\infty$-optimization for constant $\epsilon$, both of which are important and well-established problems. Though the observation that the quality of the Gaussian convolution for smoothing $\ell_\infty$ (as opposed to $\ell_2$) Lipschitz function is straightforward, that we show it leads to near-optimal $\ell_\infty$-Lipschitz convex optimization in any regime is particularly striking. Furthermore, to our knowledge the simple reduction we give from constrained to unconstrained for Lipschitz function was not explicitly known; we believe this may be useful for broader applications in convex optimization. That we make this progress by simple technical insights is not necessarily a bad thing. Finally, we think our results might shed light on how the Lovasz extension of submodular function is similar to a general $\ell_\infty$-Lipschitz function, which aids algorithm design for parallel SFM more generally.
>
>
> The reviewer also wonders about the practicality of our algorithms. Indeed, the main goal of this work is to theoretically investigate the parallel complexity of SFM. The empirical performance of our first algorithm would depend on the empirical performance of the [Carmon et al. 2023] algorithm.
>
> The reviewer also asks some interesting questions! Thank you! To answer the first question as to whether Theorem 1.1 extends to minimizing real-valued functions, indeed it does, and we will mention this in the next iteration of the paper. The second question is whether the techniques from ALS20 could speed up the algorithm. While we are not sure of its answer, the approach in ALS20 for sampling subgradients via a data structure depends on the interplay between submodular properties of the Lovasz extension and the structure of the iterative methods applied. It is not immediately clear that such an approach extends to our setting given the different methods. Nevertheless, we agree that this suggests an interesting question for future work, e.g., is it possible to decrease the query complexity up our algorithm to something with subquadratic or near-linear dependence on $n$. We may remark on this in the final version.
>
> The reviewer also points out some editorial comments; we thank the reviewer for pointing these out, and we will fix these in our next version.

---

> > ### Comment · Reviewer_xv3B · 2023-08-10
> > **Response to rebuttal**
> >
> > Thank you for your response.
> > I agree obtaining interesting results using simple technical insights is not a bad thing.
> > I will raise my score by one point, as I think the results presented in this paper will be interesting to the community, even if the proposed algorithms are not practical.

---

### Official Review · Reviewer_QKZK · 2023-07-03

**Soundness:** 4 excellent
**Presentation:** 4 excellent
**Contribution:** 3 good
**Rating:** 8
**Confidence:** 2

**Summary:**

This paper studies the parallel complexity of submodular function minimization: given a submodular function defined on the $2^n$ subsets of $[n]$ taking values between $-M$ and $M$, find a minimizing subset of $[n]$. It provides two upper bounds: one algorithm with polynomial query complexity in both $M$ and $n$ and taking $O(n^{\frac{1}{3}}M^{\frac{2}{3}})$ rounds, and the other with query complexity $n^{O(m)}$ in two rounds.

**Strengths:**

Writing is clear and convincing, with all relevant technical background and results clearly explained. Related work seems to be cited and placed in context, and open questions are identified in the conclusion. Although I am unfamiliar with the area, the problem seems fundamental and relevant to the conference and machine learning more widely.

**Weaknesses:**

My only (minor) criticism is that it took me a while to understand the difference between the "parallel" setting of SFM and the standard sequential setting. I eventually realized that the number of rounds used is essentially a measure of the *adaptivity* of an algorithm - the more adaptive an algorithm is, the more rounds it needs. I don't think the word adaptivity is mentioned in the paper, so maybe this intuition could be explained slightly better.

Perhaps a couple more sentences on the wider implications of the results for machine learning more broadly might also be nice.

typo, line 253: "then" -> "than"

**Questions:**

Could you clarify what "polynomial time" means in the context of line 26? In this model, we are only interested in query and round complexity, right?

**Limitations:**

All limitations seem to be addressed.

---

> ### Author Rebuttal · Authors · 2023-08-09
>
> We thank the reviewer for their time to carefully read and review our paper.
>
> In the weakness section, the reviewer points to the clarification between “adaptivity” versus “parallelism”; we agree that explaining this clearly is a good idea, and we will include this explanation in the next version of our paper. The reviewer also asks about the wider implications to ML. In the past decade or so, submodular function optimization has arisen quite in multiple ML applications (see https://arxiv.org/pdf/2202.00132.pdf, for instance), and with the rise of infrastructure for massively parallel computation, it is an important question to understand parallel algorithms for the same, and “low-depth” algorithms would have implications to the ML applications.
>
> In the questions section, the reviewer asks us to clarify the meaning of polynomial time, versus round and query complexity. In the context of SFM, polynomial *time* simply means that the additional computational work apart from the queries takes polynomial time; a little more formally, the total time is $O(n^c)$ additional arithmetic operations for some constant $c>0$. In the context of line 26, we are only talking about sequential algorithms, as we have not introduced the model of parallel computation yet at this point. For sequential algorithms, prior work has studied fast SFM algorithms that have both small query complexity and small additional computational cost.
>
> In the parallel complexity model, we are primarily interested in the parallel round complexity and query complexity, but ideally the algorithm’s additional computational cost (also sometimes referred to as “work” in this community) should be polynomially bounded from its query complexity (which is the case for both of our algorithms). It is true, however, that the focus of our paper was on the round and query complexity. We hope this clarifies the reviewer’s question.

---

### Official Review · Reviewer_RFBf · 2023-07-06

**Soundness:** 3 good
**Presentation:** 2 fair
**Contribution:** 3 good
**Rating:** 6
**Confidence:** 3

**Summary:**

The paper studies the parallel complexity (number of rounds of queries) for polynomial query-complexity submodular function minimization. The main result of the paper is  an $\tilde{O}(n^{1/3} M^{2/3})$ bound for this, where $n$ is the size of the ground-set and the function is integer valued with an upper bound of $M$ on the its absolute value. The approach of this work is to reduce to the problem of minimizing the Lovasz extension (which is $L = 3M$-Lipschitz) over the unit $n$-dim $\ell_\infty$-ball, and then use a simple regularization term along with the $L$-Lipschitzness to reduce it to an unconstrained minimization problem. The algorithm for this optimization is derived from the
recent work of [Carmon et al. 23] who gave a  $\tilde{O}(d^{1/3} \epsilon^{-2/3})$-round parallel $\epsilon$-approximation for $1$-Lipschitz (over $\ell_2$) convex function minimization with a minimizer in the $d$-dim unit $\ell_2$-ball and a subgradient oracle. To adapt this to the $\ell_\infty$ case, the paper first shows that a certain Gaussian convolution has better bounds with $\ell_\infty$-Lipschitzness, effectively canceling out the blowup incurred by the $\ell_\infty$ vs. $\ell_2$ norm bound. This, along with the fact that the subgradients of the Lovasz extension can be efficiently computed by queries to the function, yields the result. Additionally, the paper also gives a simple combinatorial 2-round $O(n^{M+1})$ query algorithm for this problem.


**Strengths:**

For small $M$, the main result of work improves upon previous algorithms which required $\Omega(n)$ rounds, and gives the first bound which is sublinear in n, albeit dependent on M, with the dependence on $n$ matching the lower bound of $n^{1/3}$ given by [CCK21]. In the process, the paper extends the work of [Carmon et al. 23] to the $\ell_\infty$ case essentially matching the lower bound [Nem94, DG19] in terms of the dimension. Overall the algorithmic bounds achieved by the paper are notable improvements (and optimal in some parameters) for these problems.

**Weaknesses:**

The main result of the paper is based on the recent result of [Carmon et al. 23]. The fact that SFM can be reduced via the Lovasz extension to optimization over $\ell_\infty$ is well-known and the main result of this work is to adapt the results on optimization over $\ell_2$ of  [Carmon et al. 23] to the $\ell_\infty$ case. The techniques for this are fairly straightforward. Overall, the contributions do not provide substantially novel methodology for such problems, and may not be sufficiently insightful or deep.

**Questions:**

Editorial comments:
1. Line 81: “an convex function” - - > “a convex function”.
2. Lines 258-259: $\pi_x$ should be used instead of $\pi$.
3. In Sec. 2.2 $d$ is used to denote the dimension, however $n$ is used instead in lines 302, 333.
4. Line 285 mentions “above theorems”, which is unclear. Please refer to the relevant theorems.
5. Line 311-312: refer to Fact 2.4 for the subgradient oracle $g$.
6. Line 335: “subscript” - - > “superscript”.


**Limitations:**

Yes

---

> ### Author Rebuttal · Authors · 2023-08-09
>
> We thank the reviewer for their time to carefully read and review our paper.
>
>
> In the weakness section, the reviewer mentions how it is well known that SFM can be reduced to $\ell_{\infty}$-Lipschitz optimization. We agree. However, we believe that showing that the $\ell_{\infty}$ Lipschitzness of the Lovasz extension when coupled with our new parallel convex optimization methods tailored to $\ell_{\infty}$ Lipschitz functions lead to improved parallel SFM algorithms is an important new insight. Furthermore, to our knowledge the simple reduction we give from constrained to unconstrained for Lipschitz function was not explicitly known; we believe this may be useful for broader applications in convex optimization.
>
> While we agree that our techniques are somewhat straightforward given the framework in [Carmon et al. 23], our broader contribution is the aforementioned connection we make between parallel SFM and $\ell_\infty$-optimization and the near optimal depth algorithm for $\ell_\infty$-optimization for constant $\epsilon$, these both constitute important progress on well-established open problems.  Though the observation that the quality of the Gaussian convolution for smoothing $\ell_\infty$ (as opposed to $\ell_2$) Lipschitz function is straightforward, that we show it leads to near-optimal $\ell_\infty$-Lipschitz convex optimization in any regime is particularly striking. That we make this progress by simple technical insights is not necessarily a bad thing.
>
> In the Questions section, the reviewer suggests many editorial comments. We agree with all of them, and we will incorporate these changes in the next version. Thank you!

---

> > ### Comment · Reviewer_RFBf · 2023-08-16
> >
> > Thank you for the response, this mostly addresses my concerns. Please add the explanations provided in the rebuttal to the paper as well. I have raised my rating by one.

---

### Official Review · Reviewer_T5ni · 2023-07-08

**Soundness:** 4 excellent
**Presentation:** 4 excellent
**Contribution:** 4 excellent
**Rating:** 7
**Confidence:** 3

**Summary:**

This paper considers the submodular minimization problem (SFM), where the submodular function $f$ is bounded between $-M$ and $M$. There has been a large number of papers that focus on solving this problem in as few queries as possible, but the proposed algorithms are generally highly sequential (at least $\Omega(n)$). On the other hand, there is a lot of work on the parallel complexity of SFM. The contribution of this paper is to propose and analyze algorithms with low parallel complexity.

They first propose an algorithm for SFM that runs in $\tilde{O}(n^{1/3}M^{2/3})$ rounds. This algorithm uses a reduction from parallel SFM to parallel convex optimization in order to achieve this. They provide an improved method for parallel convex optimization to be used for their algorithm. Next, they propose an algorithm that is just 2 rounds but with query complexity $n^{O(M)}$, which is optimal if $M$ is constant. This second algorithm is very simple and clean.


**Strengths:**

- The problem setting is very well motivated since SFM includes many applications that involve large datasets and therefore parallel algorithms are needed. This paper also fills an interesting gap in existing literature on SFM, where there is work on the parallel complexity but no algorithms that run in few rounds.
- Very strong theoretical results and ideas, which could be useful for other problems.
- Extremely clear and well-written.

**Weaknesses:**

- There is no experimental evaluation, but this paper is a theory paper so it is not much of an issue.

**Questions:**

None

**Limitations:**

Yes

---

> ### Author Rebuttal · Authors · 2023-08-09
>
> We thank the reviewer for a careful reading of our paper and writing the review.
>
> In the weakness section, the reviewer points to the lack of experiments. As the reviewer also notes, the main focus of the paper is exploring algorithms for SFM and $\ell_{\infty}$ convex optimization through a theoretical lens.

---

### Official Review · Reviewer_dKXF · 2023-07-25

**Soundness:** 4 excellent
**Presentation:** 4 excellent
**Contribution:** 3 good
**Rating:** 7
**Confidence:** 4

**Summary:**

	In this paper, the author(s) investigate the classic submodular minimization problem within the parallel computing regime. The setting is an integral submodular function $f$ defined on $2^{[n]}$ which is bounded ($|f| \le M$) and normalized ($f(\emptyset) = 0$. There primary contribution is obtaining query-efficient $M$-dependent algorithms with sublinear in $n$ rounds. This is achieved by regularizing the Lov{\'a}sz extension of the submodular function, and then using some prior results on parallel convex optimization. They also proving a 2-round $O(n^{M+1})$-query algorithm for submodular minimization.

**Strengths:**

The motivations and background to the results are clearly stated. The proof ideas are clearly written.

**Weaknesses:**

	Page 2, line 69 capitalize section 2.1.

	Page 2, line 70 minmizer misspelled.

	Page 3, line 114 capitalize theorem 1.1.

	Page 5, line 225 comma needed after "As discussed in the introduction.

	Page 6, Lemma 2.3 any $c,x \in \mathbb{R}^n$ I believe should be $c,r \in \mathbb{R}^n$.

	Page 9, line 359 in Algorithm 1 line 4, computing $A(S)$ may require an input of size $M + 1$, i.e., when $|S| = M$. The proof on this line only requires inputs of size $M$.

	Page 9 line 378, $f(S_* \cup j)$ change to $f(S_* \cup \{j\})$.

**Questions:**

I have no further questions then the potential confusions stated in the weaknesses section.

**Limitations:**

The author(s) clearly state the assumptions needed for their algorithms and end with some potential open questions on how their work may be improved.

---

> ### Author Rebuttal · Authors · 2023-08-09
>
> We thank the reviewer for a careful reading of our paper and writing the review.
>
> In the weakness section, the reviewer points out several typographical errors; we agree with them all, and will fix them in the next version. For line 359 specifically, indeed what we should have stated is that the size of all sets queried in Line 4 is at most $M+1$ not $M$ as initially stated. This still allows for the algorithm to be implemented in 2 rounds and does not change the query complexity of our algorithm.
>
> Thank you!

---

### Decision · Program_Chairs · 2023-09-21

**Decision:**

Accept (spotlight)

**Comment:**

This work obtains new query-efficient o(n)-round M-dependent algorithms for submodular function minimization.